

# Unprofessional peer reviews disproportionately harm underrepresented groups in STEM

Nyssa J. Silbiger[1],* and Amber D. Stubler[2],*

[1] Biology Department, California State University, Northridge, CA, USA
[2] Biology Department, Occidental College, Los Angeles, CA, USA
* These authors contributed equally to this work.

## ABSTRACT

**Background:** Peer reviewed research is paramount to the advancement of science. Ideally, the peer review process is an unbiased, fair assessment of the scientific merit and credibility of a study; however, well-documented biases arise in all methods of peer review. Systemic biases have been shown to directly impact the outcomes of peer review, yet little is known about the downstream impacts of unprofessional reviewer comments that are shared with authors.

**Methods:** In an anonymous survey of international participants in science, technology, engineering, and mathematics (STEM) fields, we investigated the pervasiveness and author perceptions of long-term implications of receiving of unprofessional comments. Specifically, we assessed authors' perceptions of scientific aptitude, productivity, and career trajectory after receiving an unprofessional peer review.

**Results:** We show that survey respondents across four intersecting categories of gender and race/ethnicity received unprofessional peer review comments equally. However, traditionally underrepresented groups in STEM fields were most likely to perceive negative impacts on scientific aptitude, productivity, and career advancement after receiving an unprofessional peer review.

**Discussion:** Studies show that a negative perception of aptitude leads to lowered self-confidence, short-term disruptions in success and productivity and delays in career advancement. Therefore, our results indicate that unprofessional reviews likely have and will continue to perpetuate the gap in STEM fields for traditionally underrepresented groups in the sciences.

Corresponding author
Nyssa J. Silbiger,
nyssa.silbiger@csun.edu

## INTRODUCTION

The peer review process is an essential step in protecting the quality and integrity of scientific publications, yet there are many issues that threaten the impartiality of peer review and undermine both the science and the scientists (*Kaatz, Gutierrez & Carnes, 2014*; *Lee et al., 2013*). A growing body of quantitative evidence shows violations of objectivity and bias in the peer review process for reasons based on author attributes (e.g., language, institutional affiliation, nationality, etc.), author identity (e.g., gender, sexuality) and reviewer perceptions of the field (e.g., territoriality within field, personal gripes with

authors, scientific dogma, discontent/distrust of methodological advances) (*Lee et al., 2013*). The most influential demonstrations of systemic biases within the peer review system have relied on experimental manipulation of author identity or attributes (e.g., *Goldberg's, 1968* classic study "Joan" vs "John"; *Goldberg, 1968*; *Wennerås & Wold, 1997*) or analyses of journal-reported metrics such as number of papers submitted, acceptance rates, length of time spent in review and reviewer scores (*Fox, Burns & Meyer, 2016*; *Fox & Paine, 2019*; *Helmer et al., 2017*; *Lerback & Hanson, 2017*). These studies have focused largely on the inequality of outcomes resulting from inequities in the peer review process. While these studies have been invaluable for uncovering trends and patterns and increasing awareness of existing biases, they do not specifically assess the content of the reviews (*Resnik, Gutierrez-Ford & Peddada, 2008*), the downstream effects that unfair, biased and *ad hominem* comments may have on authors and how these reviewer comments may perpetuate representation gaps in science, technology, engineering, and mathematics (STEM) fields.

In the traditional peer review process, the content, tone and thoroughness of a manuscript review is the sole responsibility of the reviewer (the identity of whom is often protected by anonymity), yet the contextualization and distribution of reviews to authors is performed by the assigned (handling) editor at the journal to which the paper was submitted. In this tiered system, journal editors are largely considered responsible for policing reviewer comments and are colloquially referred to as the "gatekeepers" of peer review. Both reviewers and editors are under considerable time pressures to move manuscripts through peer review, often lack compensation commensurate with time invested, experience heavy workloads and are subject to inherent biases of their own, which may translate into irrelevant and otherwise unprofessional comments being first written and then passed along to authors (*Resnik & Elmore, 2016*; *Resnik, Gutierrez-Ford & Peddada, 2008*).

We surveyed STEM scientists that have submitted manuscripts to a peer-reviewed journal as first author to understand the impacts of receiving unprofessional peer review comments on the perception of scientific aptitude (confidence as a scientist), productivity (publications per year) and career advancement (ability to advance within the field). This study defined an unprofessional peer review comment as any statement that is unethical or irrelevant to the nature of the work; this includes comments that: (1) lack constructive criticism, (2) are directed at the author(s) rather than the nature or quality of the work, (3) use personal opinions of the author(s)/work rather than evidence-based criticism, or (4) are "mean-spirited" or cruel (e.g., of comments received by survey respondents that fit these criteria see Fig. 1). The above definition was provided as a guideline for survey respondents to separate frank, constructive and even harsh reviews from those that are blatantly inappropriate or irrelevant. Specifically, this study aimed to understand the content of the unprofessional peer reviews, the frequency at which they are received and the subsequent impacts on the recipient's perception of their abilities. Given that psychological studies show that overly harsh criticisms can lead to diminished success (*Baron, 1988*), we tested for the effects of unprofessional peer review on the perception of scientific aptitude, productivity and career advancement of all respondents.

"The author's status as a trans person has distorted his view of sex beyond the biological reality."

"Despite being a woman, the PI was trained by several leading men in the field and is thus likely adequately prepared to lead the proposed research."

"The first author is a woman. She should be in the kitchen, not writing papers"

"The author's last name sounds Spanish. I didn't read the manuscript because I'm sure it's full of bad English"

"This paper is, simply, manure"

"The authors study design setback the advancement of the field by 20 years"

"I said that I'd never again cite or review a paper written by [XX] so it pains me to learn that this is one of their students. God help them."

"What the authors have done is an insult to science"

"You should look closely at a career outside of science."

"[X] tried this in the 1990s and failed and he was more creative than you".

"The phrases I have so far avoided using in this review are, "lipstick on a pig", and "bullshit baffles brains.""

"In short, this piece of research bears all the hallmarks of some bright people who saw an opportunity in a currently hot field of research, and thought they would jump in because, after all, how hard could it be? I have scanned the resumes of every one of the authors, and have come to the conclusion that they are indeed very bright people who could have used some good advice before starting this. The passage of this manuscript would have been much easier, and I would not have had to work so hard"

"This is obviously written by a group from a lower standardized institution based on the quality of work."

"This person works for an NGO, you shouldn't believe anything they say."

**Figure 1 Examples of unprofessional peer reviews from survey respondents.** Permission to publish these comments was explicitly given by respondents who certified the comments were reported accurately.

Underrepresented groups in particular are vulnerable to stereotype threat, wherein negative societal stereotypes about performance abilities and aptitude are internalized and subsequently expressed (*Leslie et al., 2015*). Further, the combination of social categorizations may lead to amplified sources of oppression and stereotype threat (*Crenshaw, 1991*); therefore, it is necessary to assess the impacts of unprofessional peer review comments across intersectional gender and racial/ethnic groups.

## MATERIALS AND METHODS

### Survey methods and administration

The data for this study came from an anonymous survey of international members of the STEM community using Qualtrics survey software. Data were collected under institutional review board agreements and federalwide assurance at Occidental College (IRB00009103, FWA00005302) and California State University, Northridge (IRB00001788, FWA00001335). The survey was administered between 28 February 2019 and 10 May 2019 using the online-based platform in English and was open to anyone self-identifying as a member of the STEM community that published their scholarly work in a peer reviewed system. Participation in the survey was voluntary and no compensation or incentives were offered. All respondents had to certify that they were 18 years or older and read the informed consent statement; participants were able to exit the survey at any point without consequence. Participants were recruited broadly through

social media platforms, direct posting on scientific list-serves and email invitations to colleagues, department chairs and organizations focused on diversity and inclusive practices in STEM fields (see Supplemental Files for distribution methods). Targeted emails were used to increase representation of respondents. Data on response rates from specific platforms were not collected.

The survey required participants to provide basic demographic information including gender identity, level of education, career stage, country of residence, field of expertise and racial and/or ethnic identities (see Supplemental Files for specific survey questions). Throughout the entire survey, all response fields included "*I prefer not to say*" as an opt-out choice. Once demographic information was collected from participants, the study's full definition of an unprofessional peer review was presented and respondents self-identified whether they had ever received a peer review comment as first author that fit this definition. Survey respondents answering "*no*" or "*I prefer not to say*" were automatically redirected to the end of the survey, as no additional questions were necessary. Respondents answering "*yes*" were asked a series of follow-up questions designed to determine the nature of the unprofessional comments, the total number of scholarly publications to date and the number of independent times an unprofessional review was experienced.

The perceived impact of the unprofessional reviews on the scientific aptitude, productivity and career advancement of each respondent was assessed using the following questions: (1) To what degree did the unprofessional peer review(s) make you doubt your scientific aptitude? (1–5) 1 = not at all, 5 = I fully doubted my scientific abilities; (2) To what degree do you feel that receiving the unprofessional review(s) limited your overall productivity? Please rate this from 1 to 5. 1 = not at all, 5 = greatly limited number of publications per year; (3) To what degree do you feel that receiving the unprofessional review(s) delayed career advancement? Please rate this from 1 to 5. 1 = not at all, 5 = greatly impacted/delayed career advancement. Finally, respondents were invited to provide direct quotes of unprofessional reviews received (although the respondents were told they could remove any personal or identifying information such as field of study, pronouns, etc.). Participants choosing to share peer review quotes were able to specify whether they gave permission for the quote to be shared/distributed. Explicit permission from each respondent was received to use and distribute all quotes displayed in Fig. 1. At the end of the survey, all respondents were required to certify that all the information provided was true and accurate to the best of their abilities and that the information was shared willingly.

We recognize that there are limitations to our survey design. First, our survey was only administered in English. There also may have been non-response bias for individuals who did not experience negative comments during peer review even though any advertisement of this survey indicated that we sought input from anyone who has ever published a peer-reviewed study as first author. There could also be a temporal element, where authors who received comments more recently may have responded differently than those who received unprofessional reviews many years ago. Additionally, the order of questions was not randomized and participants were asked to complete demographic information before answering questions about their peer review experience, which may have primed respondents to select answers that were more in line with racial or gender-specific

stereotypes (*Steele & Ambady, 2006*). In order to maintain the anonymity of our respondents, we did not ask for any bibliometric data from the authors. Given that our sample of respondents represented a diverse array of career stages, STEM fields, countries of residence and racial/ethnic identities, we do not believe that any of the above significantly limits the interpretation of our results.

## Data analysis

We tested for the pervasiveness and downstream effects of unprofessional peer reviews on four intersecting gender and racial/ethnic groups: (1) women of color and non-binary people of color, (2) men of color, (3) white women and white non-binary people and (4) white men. Due to the small number of respondents identifying as non-binary (<1% of respondents), we statistically analyzed women and non-binary genders together in a category as marginalized genders in the sciences. However, refer to Table S1 for full breakdown of responses from each gender identity so that patterns may be assessed by readers without the constraints conferred by statistical assumptions and analyses.

To protect the anonymity of individuals that may be personally identified based on the combination of their race/ethnicity and gender, respondents who identified as a race or ethnicity other than white, including those who checked multiple racial and/or ethnic categories were grouped together for the statistical analysis (Fig. S1). It is important to note that by grouping the respondents into four categories, the analysis captures only the broad patterns for intersectional groups and does not relay the unique experiences of each respondent, which should not be discounted.

Survey respondents ($N$ = 1,106) were given the opportunity to opt out of any question; therefore, the sample sizes were different for each statistical analysis. We tested for differences in the probability of receiving an unprofessional peer review across four intersectional groups ($N$ = 1,077) using a Bayesian logistic regression (family = Bernoulli, link = logit). Of the 642 people who indicated that they received an unprofessional peer review, 617, 620 and 618 answered the questions regarding perceived impacts to their scientific aptitude, productivity and career advancement, respectively. We ran individual Bayesian ordinal logistic regressions (family = cumulative, link = logit) for each of the three questions to test for differences in probabilities of selecting a 1–5 across the four groups. All models were run using the BRMS package (*Bürkner & Others, 2017*) in R*v3.5.2* which uses the Hamiltonian Monte Carlo algorithm in STAN (*Hoffman & Gelman, 2014*; *Stan Development Team, 2015*). Each model was sampled from four chains, 4,000 iterations post-warmup, and half student $t$ distributions for all priors. Model convergence was assessed using Gelman–Rubin diagnostics ($\hat{R}$ < 1 for all parameters; *Gelman & Rubin, 1992*) and visual inspection of trace plots. Posterior predictive checks were visually inspected using the *pp_check()* function and all assumptions were met. Data are presented as medians and two-tailed 95% Bayesian credible intervals (BCI).

## RESULTS

We received 1,106 responses from people in 46 different countries across >14 STEM disciplines (Fig. 2). Overall, 58% of all the respondents ($N$ = 642) indicated that they had
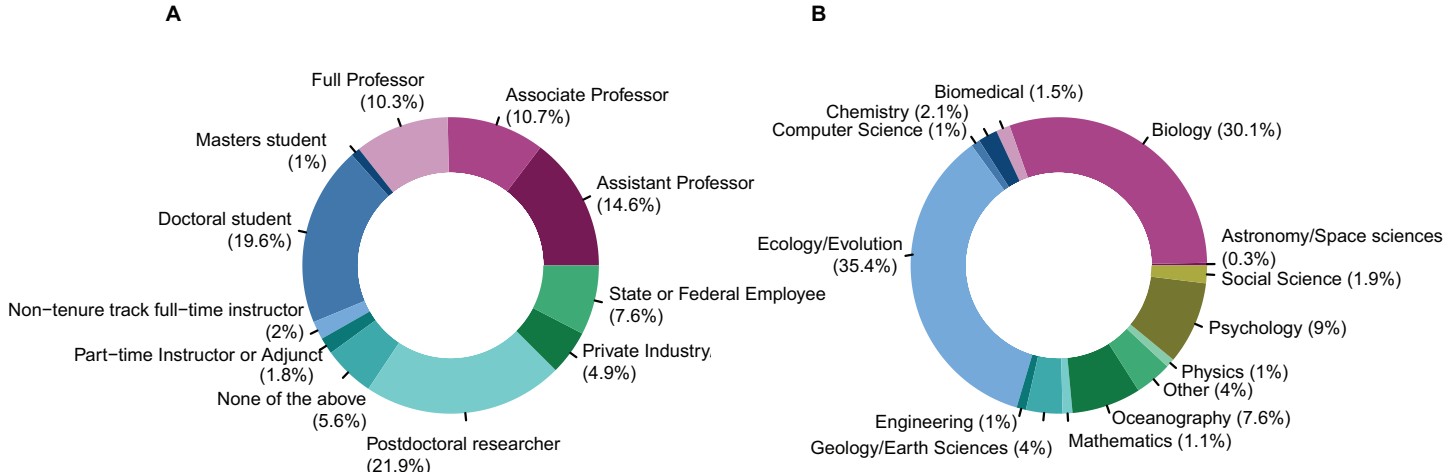

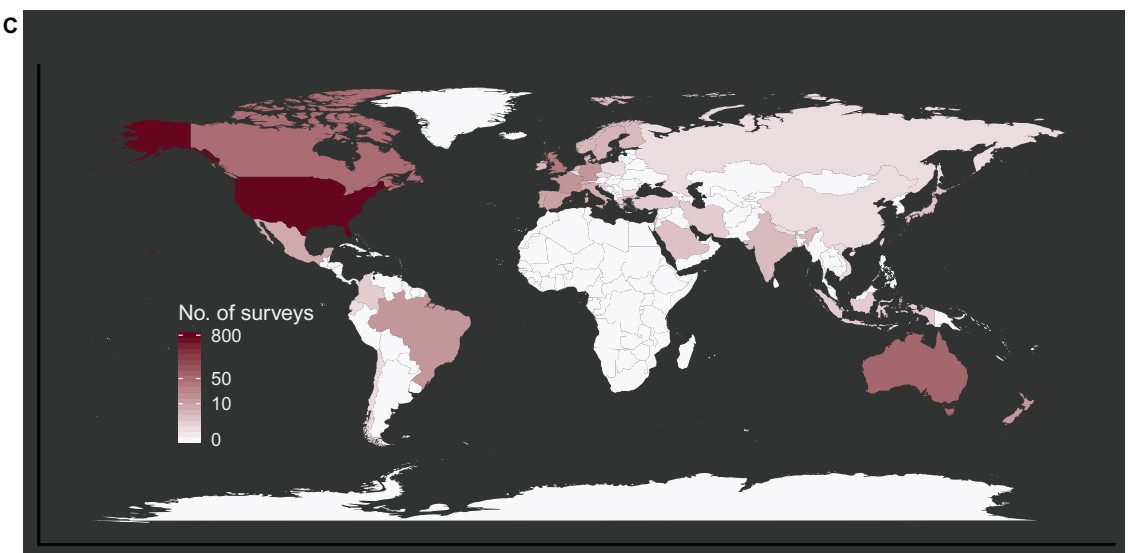

**Figure 2** **Survey demographics.** (A) Representative career stages (*N* = 11), (B) scientific disciplines (*N* = 14) and (C) countries (*N* = 46) from survey participants. Color in subset (C) represents number of surveys from each country where white is 0.

received an unprofessional review, with 70% of those individuals reporting multiple instances (3.5 ± 5.8 reviews, mean ± SD, across all participants). There were no significant differences in the likelihood of receiving an unprofessional review among the intersectional groups (Fig. S2); however, there were clear and consistent differences in downstream effects between groups in perceived impacts on self-confidence, productivity and career trajectories after receiving an unprofessional review.

White men were most likely to report no impact to their scientific aptitude (score of 1) after receiving an unprofessional peer review (P[1] = 0.40, 95% BCI [0.34–0.47], where P[score] denotes the probability of selecting a particular score), with a 5.7 times higher probability of selecting a 1 than a 5 (fully doubted their scientific aptitude; P[5] = 0.07, 95% BCI [0.05–0.09]). Notably, white men were 1.3, 2.0 and 1.7 times more likely to indicate no

resultant doubt of their scientific aptitude than men of color ($P[1] = 0.30$, 95% BCI [0.20–0.41]), white women and white non-binary people ($P[1] = 0.20$, 95% BCI [0.16–0.23]) and women of color and non-binary people of color ($P[1] = 0.23$, 95% BCI [0.16–0.31]), respectively (Fig. 3A). Together, these results indicate that receiving unprofessional peer reviews had less of an overall impact on the scientific aptitude of white men relative to the remaining three groups.

Similar patterns among intersectional groups emerged for reported impacts of unprofessional reviews on productivity (measured in number of publications per year). Specifically, women of color and non-binary people of color, white women and white non-binary people and men of color were mostly likely to select a 3 (moderate level of perceived negative impact on productivity), whereas white men were most likely to select a 1 (no perceived impact on their productivity; Fig. 3B). White men were also the least likely of all groups to indicate that receiving unprofessional reviews greatly limited their number of publications per year ($P[5] = 0.06$, 95% BCI [0.05–0.09]), which significantly differed from groups of women and non-binary people, but not men of color (Fig. 3B).

Women of color and non-binary people of color had the most distinct pattern in reported negative impacts on career advancement (Fig. 3C). Women of color and non-binary people of color had a nearly equal probability of reporting each level of impact (1–5); whereas, men of color, white women and white non-binary people and white men had a decreasing probability of selecting scores indicative of a higher negative impact on career advancement (Fig. 3C). Specifically, women of color and non-binary people of color were the most likely to select that they had significant delays in career advancement as a result of receiving an unprofessional review ($P[5] = 0.20$, 95% BCI [0.13–0.28]). Women of color and non-binary people of color were also the least likely of the groups to report no impact on career advancement as a result of unprofessional reviews ($P[1] = 0.22$, 95% BCI [0.15–0.31]).

## DISCUSSION

Our data show that unprofessional peer reviews are pervasive in STEM disciplines, regardless of race/ethnicity or gender, with over half of participants reporting that they had received unprofessional comments. Our study did not assess peer review outcomes of participants, but it is possible that unprofessional reviews could impact acceptance rates across groups differently because reviewer perception of competence is implicitly linked to gender regardless of content (Goldberg, 1968; Kaatz, Gutierrez & Carnes, 2014; Wennerås & Wold, 1997). Previous studies have demonstrated clear differences in acceptance/rejection rates between genders (Murray et al., 2019; Symonds et al., 2006; Fox & Paine, 2019) and future studies should test if receiving an unprofessional peer review leads to different acceptance outcomes depending on gender and/or race/ethnicity. While there were no statistical differences in the number of unprofessional reviews received among the four intersectional groups in our study, there were clear and consistent differences in the downstream impacts that the unprofessional reviews had among groups.

Overall, white men were the least likely to question their scientific aptitude, or report delays in productivity or career advancement than any other group after receiving an
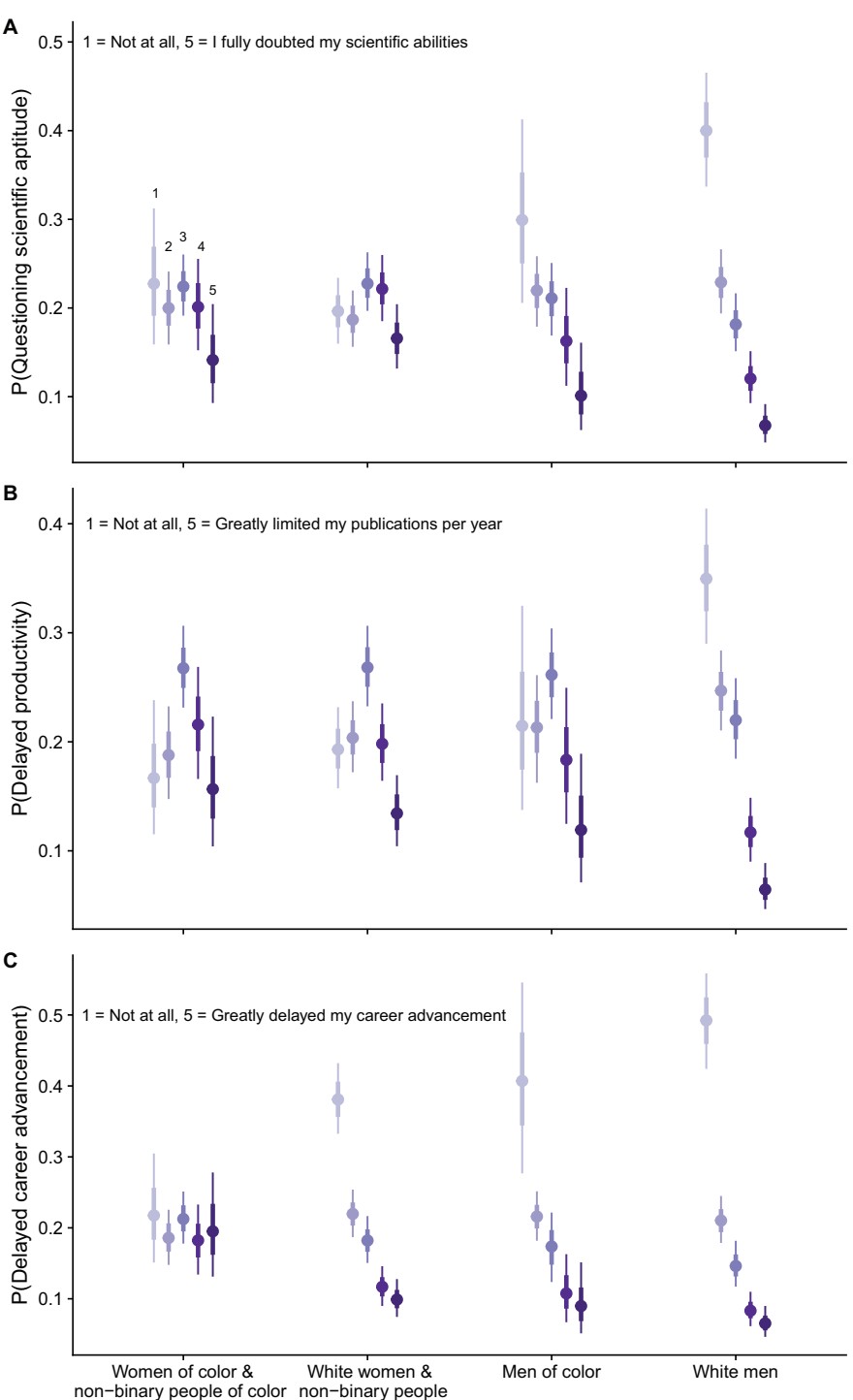

**Figure 3 Results from Bayesian ordinal logistic regression.** Figure shows the probability of selecting a 1–5 for (A) doubting scientific aptitude ($N = 617$), (B) delayed productivity ($N = 620$) and (C) delayed career advancement ($N = 618$) across intersectional groups after receiving an unprofessional peer review. Data are medians and two-tailed 95% BCI. Colors represent level of impact with the lightest (1) as no perceived impact and the darkest (5) as the highest impact. Women and non-binary people were grouped for the statistical analysis to represent marginalized genders in STEM fields.

unprofessional review. Groups that reported the highest self-doubt after unprofessional comments also reported the highest delays in productivity. This finding corroborates studies showing destructive criticism leads to self-doubt (*Baron, 1988*) and vulnerability to stereotype threat (*Leslie et al., 2015*), which has quantifiable negative impacts on productivity (*Kahn & Scott, 1997*) and career advancement (*Howe-Walsh & Turnbull, 2014*). Conversely, high self-confidence is related to increased persistence after failure (*Baumeister et al., 2003*). Therefore, scientists with a higher evaluation of their own scientific aptitude after an unprofessional review may be less likely to have reduced productivity following a negative peer review experience.

Women and non-binary people were the most likely to report significant delays in productivity after receiving unprofessional reviews. It is well known that publication rates in STEM fields differ between genders (*Symonds et al., 2006*; *Bird, 2011*). Men have 40% more publications than women on average, with women taking up to 2.5 times as long to achieve the same output rate as men in some fields (*Symonds et al., 2006*). While our study cannot confer causality leading to diminished productivity, the results show that unprofessional reviews reinforce bias that is already being encountered by underrepresented groups on a daily basis. Other well-studied mechanisms leading to reduced productivity for women include (but are not limited to) papers by women authors spend more time in review than papers by men (*Hengel, 2017*), men are significantly less likely to publish coauthored papers with women than with other men (*Salerno et al., 2019*), women receive less research funding than men in some countries (*Witteman et al., 2019*) and women spend more time doing service work than men at academic institutions (*Guarino & Borden, 2017*). Women are also underrepresented in the peer review process leading to substantial biases in peer review (*Goldberg, 1968*; *Kaatz, Gutierrez & Carnes, 2014*). For example, studies have shown that women are underrepresented as editors which leads to fewer refereed papers by women (*Fox, Burns & Meyer, 2016*; *Lerback & Hanson, 2017*; *Helmer et al., 2017*; *Cho et al., 2014*) and that authors of all genders are less likely to recommend women reviewers (*Lerback & Hanson, 2017*) which contributes to inequity in peer review outcomes (i.e., fewer first and last authored papers accepted by women) (*Murray et al., 2019*). However, some strategies, such as double-blind reviewing and open peer review, have been shown to alleviate gender inequity in publishing (*Darling, 2015*; *Budden et al., 2008*; *Groves, 2010*), although there are notable exceptions (*Webb, O'Hara & Freckleton, 2008*). The difficulty in publishing and lack of productivity may contribute to the high attrition rate of women in academia (*Cameron, Gray & White, 2013*).

Assessment of intersectional groups has been generally overlooked in research on publication and peer review biases. Yet, traditionally underrepresented racial and ethnic groups experience substantial pressures and limitations to inclusion in STEM fields. Indeed, in our study there were significant differences, especially in perceived delays in career advancement, between white women and white non-binary people, and women of color and non-binary people of color (Fig. 3C). Had we focused on only gender or racial differences, the distinct experiences of women of color and non-binary people of color would have been obscured. Because both gender and racial biases lead to diminished

recruitment and retention, as well as higher rates of attrition in the sciences (*Xu, 2008*; *Alfred, Ray & Johnson, 2019*), intersectionality cannot be ignored. Our results indicate that receiving unprofessional peer reviews is an yet another barrier to equity in career trajectories for women of color and non-binary people of color, in addition to the quality of mentorship, intimidation and harassment, lack of representation and many others (*Howe-Walsh & Turnbull, 2014*; *Zambrana et al., 2015*).

Our study indicates that unprofessionalism in reviewer comments is pervasive in STEM fields. Although we found clear patterns indicating that unprofessional peer reviewer comments had a stronger negative impact on underrepresented intersectional groups in STEM, all groups had at least some members reporting the highest level of impact in every category. This unprofessional behavior often occurs under the cloak of anonymity and is being perpetuated by the scientific community upon its members. Comments like several received by participants in our study (see Fig. 1) have no place in the peer review process. Interestingly, less than 3% of our participants that received an unprofessional peer review stated that the review was from an open review journal, where the peer reviews and responses from authors are published with the final manuscript (*Pulverer, 2010*). While a recent laboratory study showed that open peer review practices led to higher cooperation and peer review accuracy (*Leek, Taub & Pineda, 2011*), less is known about how transparent review practices affect professionalism in peer review comments. Our data indicate that open reviews may help curtail unprofessional comments, but more research on this topic is needed.

Individual scientists have the power and responsibility to address the occurrence of unprofessional peer reviews directly and enact immediate change. We therefore recommend the following: (1) Make peer review mentorship an active part of student and peer learning. For example, departments and scientific agencies should hold workshops on peer review ethics. (2) Follow previously published best practices in peer review (*Huh & Sun, 2008*; *Kaatz, Gutierrez & Carnes, 2014*). (3) Practice self-awareness and interrogate whether comments are constructive and impartial (additionally, set aside enough time to review thoroughly, assess relevance and re-read any comments). (4) Encourage journals that do not already have explicit guidelines for the review process to create a guide, as well as implement a process to reprimand or remove reviewers that are acting in an unprofessional manner. For example, the journal could contact the reviewer's department chair or senior associate if they submit an unprofessional review. (5) Societies should add acceptable peer review practices to their code of conduct and a structure that reprimands or removes society members that submit unprofessional peer reviews. (6) Editors should be vigilant in preventing unprofessional reviews from reaching authors directly and follow published best practices (*D'Andrea & O'Dwyer, 2017*; *Resnik & Elmore, 2016*). (7) When in doubt use the "golden rule" (review others as you wish to be reviewed).

## CONCLUSIONS

Our study shows that unprofessional peer reviews are pervasive and that they disproportionately harm underrepresented groups in STEM. Specifically,

underrepresented groups were most likely to report direct negative impacts on their scientific aptitude, productivity and career advancement after receiving an unprofessional peer review. While it was beyond the scope of this study, future investigations should also focus on the effect of unprofessional peer reviews on first-generation scientists English as a second language, career stage, peer review in grants, and other factors that could lead to differences in downstream effects. Unprofessional peer reviews have no place in the scientific process and individual scientists have the power and responsibility to enact immediate change. However, we recognize and applaud those reviewers and editors (and there are many!) that spend a significant amount of time and effort writing thoughtful, constructive, and detailed criticisms that are integral to moving science forward.

## ACKNOWLEDGEMENTS

We thank the hundreds of respondents who bravely shared some of their harshest, darkest critiques. We thank M. Siple, A. Mattheis, M. DeBiasse, J. Carroll, and the editor and two reviewers for comments and feedback on the manuscript. This is CSUN Marine Biology contribution # 296. This is a co-first authored paper with author order determined by ranking the most unprofessional peer reviews received by each author to date, and alphabetical order.

### Funding

The authors received no funding for this work.

### Competing Interests

The authors declare that they have no competing interests.

### Author Contributions

- Nyssa J. Silbiger conceived and designed the experiments, performed the experiments, analyzed the data, contributed reagents/materials/analysis tools, prepared figures and/or tables, authored or reviewed drafts of the paper, approved the final draft.
- Amber D. Stubler conceived and designed the experiments, performed the experiments, contributed reagents/materials/analysis tools, prepared figures and/or tables, authored or reviewed drafts of the paper, approved the final draft.

### Human Ethics

The following information was supplied relating to ethical approvals (i.e., approving body and any reference numbers):

Occidental College (IRB00009103, FWA00005302) and California State University, Northridge (IRB00001788, FWA00001335) granted ethical approval to carry out the study.

### Data Availability

Code is available at GitHub: https://github.com/njsilbiger/UnproReviewsInSTEM.

Code is also available at Zenodo: DOI 10.5281/zenodo.3533928.

The survey collected no unique personal identifying information beyond basic demographic information. However, respondent anonymity may be jeopardized through revealing specific combinations of the demographic information collected (e.g., field of study, gender, race/ethnicity). The data that support the findings of this study have been redacted to protect the anonymity of the respondents and is available as a Supplemental File.

## Supplemental Information

Supplemental information for this article can be found online at http://dx.doi.org/10.7717/peerj.8247#supplemental-information.

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
