# Peer review of "Unprofessional peer reviews disproportionately harm underrepresented groups in STEM"

_PeerJ, doi:10.7717/peerj.8247_

## Round 0.1 · original submission · Minor Revisions

We now have comments back from 2 reviewers who are both experienced and are knowledgeable about some of the shortcomings of the peer-review process because they have themselves published on some of these issues. Both were highly supportive of your work and recommended acceptance, and my own reading of the manuscript is also positive. One of the referees felt that it was excellent and recommended the paper be accepted as is, while the other provides a thoughtful list of suggestions for feedback on your paper for ways in which the manuscript could be improved through clarifications or additional information.
I expect these suggestions will be relatively straightforward for you to address, and I look forward to seeing your revised manuscript. Please do not hesitate to contact me if you have any questions or need any clarification to aid with your revisions.

·

Basic reporting

The article meets the basic reporting criteria very well.

Experimental design

Experimental design is excellent.

Validity of the findings

Findings are valid.

Additional comments

This is a very important and useful survey. It provides empirical evidence for what this reviewer has suspected for many years--that unprofessional reviews can negatively impact career advancement and discourage researchers. It affirms the importance of training researchers on how to do professional reviews and for editors to pay close attention to peer reviewer comments and edit them.

·

Basic reporting

Please see below. I would suggest adding the entire survey in an appendix. This is an exploratory study so no specific hypotheses were given.

Experimental design

Please see feedback below. The research is appropriate to the scope of the journal.

Validity of the findings

The findings are novel. I cannot comment on statistical validity or reporting.

Additional comments

The manuscript aims to assess the impact of unethical peer review among STEM participants specifically capturing participants’ perceptions of scientific aptitude, overall productivity and delays in career advancement.

The authors have done a terrific job clearly outlining the gap in the literature surrounding the ethics of peer review, namely that there is little known about the downstream impact of unprofessional reviews on the researchers. Therefore, the findings reported here are of great value to researchers interested in research integrity and I applaud the researchers for undertaking this study. I have outlined several areas that I believe could benefit from greater clarity or explanation which I hope serves to strengthen the manuscript.

The introduction reads well and sets the stage nicely for undertaking the study. Certainly it could be further expanded but it is well written, clear and concise. I have no comments here.

Materials and Methods:

1) Thank you for doing a nice job explaining the ethics and the protections to research participants. I am interested in knowing more about the recruitment strategy including its rationale. Why were social media platforms used, especially as there is some data suggesting that not all academics in all positions use social media uniformly. Why twitter and Facebook specifically? Which listservs and how was it decided who to invite by email. The current recruitment method seems very unsystematic. In addition, there are no details on how contact was initiated, e.g., was recruitment done to a specific Facebook group(s)? And if so why? Were secondary emails sent to invitees (basically was there any follow-up to initial recruitment)? Details about recruitment are important not only for readers to assess the rigor of the methods, but also to reproduce the research.

2) There seems to be no information pertaining to how many people were contacted and what was the response rate. Is this because the researchers didn’t start with a known number of potential participants (because of recruitment through social media) and so the denominator remains unknown? Please explain.

3) While I understand the brief rationale given in the Introduction surrounding why specifically scientific aptitude, productivity and career advancement were chosen, I am unsure about how these items and scales were designed. I presume these are not borrowed scales and no such scales could be identified. I would explain this. So how were these items designed? Was this based on the existing literature, was any pretesting of questions done using interviews prior to survey dissemination? My reason for asking for such details is that the terms “scientific aptitude,” “overall productivity,” and “career advancement” are likely to mean different things for different respondents. There is also a temporal element as certainly a bad review when I was a trainee might impact me temporarily as an Associate Professor responding today. Can more of a description be given for choosing the language used in these items? The design of this instrument might be considered a limitation.

4) Thank you for adding a limitations section. In my experience, limitations for empirical social science studies are provided in the Discussion section usually as a last paragraph before the Conclusion. This is a formatting issue so I will leave it to the authors/editors to decide where it is best placed.

5) I understand that first authors of papers were invited. How was this determined? There was no bibliometric method used to identify first authors so how was this determined? In addition, were these authors of single blinded peer review, double or open? I presume judging from the comments in Fig 1 that the respondents were part of a single blind or open review process because reviewers knew the names. If this is unknown, please state this in the paper.

6) I am a bit confused as to how the survey was conducted. When the first set of questions about whether the respondents received an unprofessional review was asked among 1106 respondents, only 642 said yes which meant the remainder only gave demographic info and no further questions were given. To me, this is a presurvey question literally as a way to screen for the inclusion of participants. So at the end of the day, you are working with 642 participants. Is this correct? If so can the method simply not be written as we prescreened participants based on whether they answered “yes” to having received an unprofessional peer review based on the screening criteria?

Results
7) There is language in the results and in the discussion that I would like clarification on. The intersectional comparisons showed no significant differences as indicated on lines 202-3 and the statement in the discussion line 246-8 “while there were no difference in the perception of receiving unprofessional comments among the four intersectional groups in our study, there were clear and consistent differences in downstream effects across groups.” These statements can be confusing to a general academic science audience who would be interested in this paper. Firstly, what the researchers are doing is to capture the perceptions of participants who have experienced unprofessional reviews so saying there were no differences in perceptions of receiving unprofessional comments among intersectional groups, but then saying there were difference in downstream effects is confusing and sounds contradicting. Please help me understand the meaning of the above sentence.

Discussion
8) Would it be possible to unpack some of the ethics surrounding unprofessional reviews? I have once in my life received an unethical review which did impact me and I was struck by the types of comments in Figure 1. In fact, I would ask the authors to consider calling these even unethical reviews. My comment is whether the authors might consider discussing further the impact of open peer review practices and how this might (or might not) curtail unethical peer reviews.

9) Another area I think is worth discussing is the role of Associate Editors/Editors when receiving unethical peer reviews. What should editors do in such cases beyond simply redacting such useless and unprofessional comments when providing feedback to the authors? As an AE for a couple of journals, I have never come across such comments from reviewers. Personally, I would consider removing the reviewer from our database completely, making it known to the senior editorial team, and would probably send an email to the reviewer personally. But this is a weak reprimand. In addition, the suggestions provided in the final paragraph of the discussion are all about ensuring practice of peer review is known to reviewers and does not touch upon how to deal with reviewers who commit such misbehaviors. While these positively framed suggestions are worthwhile to explain, positive attributes can only go so far and sometimes, reprimands are necessary to guide ethical behavior. Can the authors identify editorial responsibilities in the context of receiving unethical peer reviews? And also, can the authors perhaps consider what types of reprimands could be given to reviewers of unethical peer reviews. Should editors contact institutions and if so whom within the institution should be contacted. This paper has obvious implications for editorial teams interested in ensuring the ethical practice of peer review and editorial conduct. Greater discussion here would certainly be helpful.

---

## Round 0.2 · accepted · Accept

Both referees were positive about the initial submission, and your manuscript now incorporates all the constructive feedback provided into the revised submission. I see no reason to delay the process and am satisfied with your revisions. Therefore, I am happy to move this along into production and thank you for selecting PeerJ to publish your manuscript. I look forward to seeing this one come out.